# Smart Preventive Maintenance of Hybrid Networks and IoT Systems Using Software Sensing and Future State Prediction

**DOI:** 10.3390/s23136012

**Published:** 2023-06-28

**Authors:** Marius Minea, Viviana Laetitia Minea, Augustin Semenescu

**Affiliations:** 1Department Telematics and Electronics for Transports, University Politehnica of Bucharest, 060042 Bucharest, Romania; 2Department IT, Orange Services Romania, 020334 Bucharest, Romania; viviana.minea@stud.etti.upb.ro; 3Faculty of Materials Science and Engineering, University Politehnica of Bucharest, 060042 Bucharest, Romania; augustin.semenescu@upb.ro; 4Romanian Academy of Scientists, 050045 Bucharest, Romania

**Keywords:** preventive maintenance, Markov Chains, future state prediction, state matrix, risk assessment

## Abstract

At present, IoT and intelligent applications are developed on a large scale. However, these types of new applications require stable wireless connectivity with sensors, based on several standards of communication, such as ZigBee, LoRA, nRF, Bluetooth, or cellular (LTE, 5G, etc.). The continuous expansion of these networks and services also comes with the requirement of a stable level of service, which makes the task of maintenance operators more difficult. Therefore, in this research, an integrated solution for the management of preventive maintenance is proposed, employing software-defined sensing for hardware components, applications, and client satisfaction. A specific algorithm for monitoring the levels of services was developed, and an integrated instrument to assist the management of preventive maintenance was proposed, which are based on the network of future states prediction. A case study was also investigated for smart city applications to verify the expandability and flexibility of the approach. The purpose of this research is to improve the efficiency and response time of the preventive maintenance, helping to rapidly recover the required levels of service, thus increasing the resilience of complex systems.

## 1. Introduction

Today, the current developments in large cities are oriented towards the introduction of smart applications, reduction of environmental impact, and new services to ease day-to-day living for citizens. Continuous growth in data collection, storage, processing, and transmission led to creating heterogeneous structures for communications and data mining, which are not always compatible and/or well structured. Finally, smart mobility is also seen as one of the main solutions to reduce traffic congestions, stress, and pollution in urban areas. All these services are largely based on heterogeneous telecommunications solutions and hybrid subsystems. Standards and technologies for communication evolve on a permanent basis. For example, new standardization includes Matter, an open-source connectivity standard for smart home and Internet-of-things devices, aimed at improving compatibility and security, of which Version 1.0 of the specification was published on 4 October 2022. There is also Thread, an IPv6-based, low-power mesh networking technology for Internet-of-things (IoT) products. Hence, the evolution of network standards, technologies, and hardware is in a continuous process.

On the other hand, the more intensive use of wireless sensors and non-intrusive detection of vehicles, passengers, and/or travelers, along with the introduction of other smart-city-specific services, means that communications are also extending and becoming very heterogeneous. Maintaining the necessary level of service for such complex networks is becoming a difficult task, even though several applications and techniques help maintenance operators to track and discover malfunctions and the excessive loading of network channels or slow the response of applications. Consequently, the maintenance of such complex systems and networks is also becoming complex, making it difficult for the human operators and specific services to efficiently manage all functionalities in real-time and to ensure flawless services. Up until this moment, the different solutions for hardware, traffic, and applications monitoring are not integrated in a single platform, and standardization in this field is still at a poor level. The present work is intended to represent an additional solution for improving the response time of the SMC (Services’ Monitoring Centre), considering the continuous increase in complexity of telecommunication networks, in the context of the exploitation of smart city services. Another goal is to improve the overall application response time. From the period of the pandemic until the present day, there has been an intense emphasis on the digitalization of as many services as possible that citizens can access. Therefore, there is fierce competition between companies to offer clients the opportunity to digitally achieve their forecasted goals, assessing the services from the comfort of their home. The criterion that makes the difference and leads to customer loyalty in this situation is the desired availability of applications. Situations in which applications have a high response time provide a negative experience to users and make them re-orient towards competition. Therefore, this work is also focused on improving this aspect.
-This research was aimed at creating a platform for integrated monitoring of reliability, level of service, and client satisfaction, employing simple solutions that do not require difficult programming tasks and/or intensive computing power—a solution to collect, store, and analyze all information regarding hardware/software malfunctions and application performance. The approach assumed that intelligent agents are employed for the management of different services (e.g., specific for smart city and communication networks), which collect the relevant information regarding levels of service. The collected data were stored and used to build a state matrix, which was then employed to produce a prognosis on future states of the network and to issue early warnings for preventive maintenance. This involved the integration of intelligent agents for information collection regarding hardware and software monitoring, combined with application and client satisfaction monitoring.-A model was created for a state matrix based on collected data and building a data base for cyclic and/or event-triggered updating and analysis.-An algorithm was created for building and updating the state transition matrix based on the Markov approach. This solution was chosen to keep the necessary computing power at a low level.-Development and adaptation of the solution for client satisfaction analysis were proposed.-All these approaches were integrated into a single platform to assist the maintenance operators in early detection and warning, regarding malfunctions and network decrease in performance, also based on a risk assessment matrix.

## 2. Related Work

Studies and research were performed worldwide to enhance preventive maintenance solutions and to keep in line with the rapid development of technologies and services. The management of complex networks must begin with a deep understanding of the system architecture, based on the topics defined by the ISO network management model: fault, configuration, accounting, performance, and security managements [1]. This model provides a comprehensive means for managing the major functions of network management. Modern and heterogeneous communication networks challenge the maintenance services’ accuracy and the effective processing of big data in a real-time manner. Mobility of some wireless sensors, and/or monitored devices, also may create complex behavior of network traffic, difficult for analysis and interpretation for early detection of anomalies. In this direction, deep learning has been efficiently employed to facilitate analytics knowledge discovery in big data systems to detect hidden and complex patterns. Deep learning models are applied in network traffic monitoring and analysis.

Modern communication networks, including Cognitive Radio [2,3], as well as the reliability of the communication link between the users, are based on several Quality-of-Service (QoS) indicators, such as connection availability, channel availability, service retainability, and/or network unserviceable probability. These are evaluated under a variety of channel failure and PU arrival rates, allowing for “on-line” monitoring of the network viability. Still, these improvements depend on other, random factors, such as the channel or the receiver’s availability (which may be in different state—out of reach, busy, etc.). The authors of [2,3] conclude that another important KPI of these networks’ reliability, which should be included in the QoS study, should be the receiver’s availability. Of course, this represents a very promising advance, but not all networks are presently at this stage of development. Therefore, the methodology proposed in this work comes as a complementary service to a heterogeneous type of network, integrating QoS-related data in a solution for post-processing and forecasting of the network’s state of functionality. However, for the proposed solution, mostly fixed receivers have been considered (i.e., sensors of the smart city services), and studying the availability of the receivers may constitute, probably, a future work.

The work of S. Rezaei and X. Liu [4] presented a survey on a specific part of the models for different Deep Learning-based network traffic classifications. Aniello et al. [5] also introduced, in their study, some machine learning-based models (both unsupervised and supervised) in a scenario involving malware analysis, but they do not extend their research to malware detection.

A more in-depth analysis on network traffic analysis was performed by Conti et al. [6], which had some interesting points in considering the level of traffic at which the network is monitored and the aim of this analysis. Some non-supervised learning algorithms, such as k-means, or supervised, i.e., Random Forest, are analyzed, along with a very pertinent organization of the main KPIs in traffic monitoring, such as traffic characterization, app identification, usage study, malware detection, user action identification, OS identification, position estimation, ad fraud identification, tethering (internet sharing) information, or website fingerprinting.

Additionally, Fadlullah et al. [7] presented deep learning models and architectures for network traffic control systems, covering mainly the network infrastructural aspects.

For larger networks and big data analytics, D’Alconzo et al. [8] focused on anomaly detection and security mechanisms with the purpose of identifying and reacting in a fast manner to unpredictable events while monitoring many heterogeneous events. The authors also categorize previous research on network traffic monitoring and analysis (NTMA) that work with big data approaches. In the same domain of NTMA, the work [9] by M. Abbasi, A. Shahraki and A. Taherkordi is mentioned, which provides a comprehensive review on applications of deep learning in NTMA, analyzes the integration of deep learning and NTMA, and performs a review of DL techniques for NTMA. 

Similarly related work is given in [10,11,12]: the passive flow monitoring of hybrid network connections, usefulness of machine learning in network monitoring, and the challenges and opportunities that big data present in this direction of research.

Another interesting direction of research is focused on analyzing data traffic statistics and detecting anomalies [13]. Most of the actual methods for detecting anomalies in data traffic, especially in public networks and institutions, have been analyzed and presented in a comparative study: statistically based methods, distance-based methods, density-based methods, clustering-based methods, graph-based methods, and learning-based methods. The research concludes with the proposal of including an Anomaly Detection Module (ADM), based on a combination of the above-described technologies.

There are different, other domains where this approach is also welcomed: power grids need very accurate monitoring of operation status to ensure uninterruptible operation. One solution for this is based on random matrix theory and qualitative trend analysis [14]. The solution considers two types of elements: the variability and the overall performance of the system, ignoring the complex physical structure of the power grid and using the data generated during the operation of the power grid more effectively. On the other hand, not only natural factors may produce failures of such networks. In smart metering methods, human intervention may also be a cause of malfunction, instability, or bad operation. Data-driven fraud detection methods are analyzed in [15], comprised of AI-based supervised methods, including wide and deep neural networks and multi-data-source deep learning models, along with unsupervised methods, e.g., clustering. Complementary to these methods, vulnerabilities are analyzed from as many aspects as possible, and the researchers recommend employing lightweight privacy-preserving detection to preserve relevant data for accurate detection, as well as the use of AI-based self-learning detectors.

One other and important aspect of smart city services is the distribution of utilities. Research in the direction of improvement of water distribution normal operation and validity include some innovative solutions, such as Digital Twins, for rapidly detecting leaking and maintaining pressure control, fractal control, partitioning (pressure management areas), or multi-objective optimization, which is an approach that is based on the Gomory–Hu tree to maintain control over each segment, etc. [16].

In the same domain of energy grids management, some researchers propose hybrid data transmission networks to compensate for the missing of GSM signals in remote locations. Similar hybrid networks, based on a combination of RS485 and RF modules (nRF), according to study [17], can be successfully used in solar power parks as an alternative to GSM networks.

Hardware gear can also be a cause of a system’s or a network’s malfunction. A solution for monitoring complex hardware computing equipment could be HDD failure monitoring, which is based on self-monitoring analysis and reporting technology [18].

However, in complex distribution grids, correct operation might be corrupted via false data injection attacks (FDIAs). In [19], a novel deep neural network approach is proposed to perform simultaneously distribution system state estimation calculation (using regression) and FDIA detection.

With the increasing role of complex networks in the era of information, another problem that has been in focus in recent years is the prediction of data links related to air transports networks to improve the efficiency of transportation in complex networks of airports [20].

Regarding electricity distribution, consumption, and related policies, an arising concept and a side-effect is so-called “energy justice”, concerning the effect of introducing advanced techniques for data collection and AI-related applications in the field, which may lead to privacy infringements. In [21], it is explained that “Energy justice” is a concept that has emerged predominantly in social science research to highlight that energy related decisions, particularly as part of the energy transition, should produce just outcomes. Therefore, the authors of the study recommend that “it is important to take energy justice in consideration from an early stage in the development or design of AI techniques”.

Technologies for monitoring and the maintenance of public transformers in an energy distribution network are considered in work [22]. The aim of the research is to remotely determine public transformers load and to construct a load prediction model, based on the LSTM (Long-Short Term Memory) algorithm, to be used for detection and the accurate location of heavy overload risks in advance, therefore being a preventive maintenance technique.

Preventive maintenance has always been a priority for critical applications and industry. Therefore, many researchers are focused on finding the most appropriate solutions to improve efficiency of this aspect. Different strategies are tested, and they prove their efficacity in increasing reliability [23], such as using a logistic regression model to assess the health condition of equipment and a neural network model to estimate its failure probability, considering the scheduled workloads. Besides the industrial process, employment of intelligent agents to verify on a continuous basis the load of different components in a communication network has also been implemented. The goal is to determine the best operational status of a server in each time slot, based on Markov chain models, as well as to optimize the system’s performance, which is measured in terms of throughput [24]. However, modern communication networks now rely on optical fiber, which is immune to e.m. interferences, but the optical fiber is also part of the reliability chain, so it also needs monitoring in terms of its operational status. Therefore, there are some solutions to improve the performance of FO via integration with optical amplifier boards, able to detect optical layer events and fiber soft/hard failures with online remote management [25]. Processes increase in complexity when they are developed in cloud applications. In order to extend the preventive maintenance at this level, some researchers propose a Recurrent Neural Network (RNN)-based method to proactively predict faults, in the event of insufficient resources in fog devices, based on a conceptual LSTM and novel Computation Memory and Power (CRP) rule-based network policy [26]. For networks and systems based on sensors, some authors employ Bayesian Network Models (BNM) that can be improved via fusion-learning methodology: merging different data from sensors and metrology logs, combined with a human-in-the-loop approach for expert knowledge elicitation of the BN structure [27]. Another solution is data prediction using a v-Support Vector Regression (vSVR) algorithm [28], the latter being very useful for high network loads, such as in emergency support during festivals and large-scale activities.

Other methods for improving reliability and resilience of different systems and networks with models of operation use Least Squares Support Vector Machine (LSSVM) [27], an exponentially weighted moving average method combined with a continuous deep belief network for constructing the reliability model [28], or even intelligent solutions to prevent security breaches with a delay-based attack detection and isolation scheme (DA-DIS) [29]. For underground medium-voltage power supplying networks, a novel method for improving reliability is proposed in [30], using various machine learning classification algorithms. 

When complex systems, including more networks and subsystems, are to be monitored, different approaches include dedicated sensors, IoT platforms, and a LSTM ensemble neural, which are all developed to predict the operational status [31], and for avoiding cascading failures, a hybridization of two meta-heuristic techniques, namely, the snake optimizer and the sine-cosine algorithm (SO-SCA), are proposed to solve the problem [32]. A fault-tolerant topology algorithm for agricultural WSN, based on a double-price function, is designed in [33] to improve the connectivity and reliability of the WSN, while some approaches employ a trained multi-agent for comparing the computed future state with the actual state and early detect faults [34].

Many of the techniques applied for improving early fault detection and preventive maintenance are reviewed and analyzed together [35,36]. The authors conclude that “These monitoring tools can be used for achieving the goal of high performance and reliable networks as they are capable of analyzing the resources for configuring the network problems and alert the administrator if any network issue occurs”.

When it comes to preventive maintenance, grid networks and energy supplying distributed systems are in the center of preoccupation; methodologies include: distributed data collection network [37] or adding QoS to low cost protocols, such as ZigBee (using IEEE 802.15.4 defined physical and MAC layer) and Bluetooth (IEEE 802.15.1), by providing differentiated service for traffic of different priority at the MAC layer [38]; also, the DFS (Depth First Search) algorithm is used to divide the network in zones and to capture the influence of maintenance decisions in the model of the served load from DGs and batteries by generating topological constraints [39]. Finally, state transitions and risk models [40] have also been employed for the preventive maintenance. Regarding communication networks, different approaches are considered by several researchers: usage of infrastructure monitoring tools [41], cloud applications monitoring [42], runtime software-fault monitoring tools [43], distributed performance monitoring [44], or lightweight distributed metric services [45] to cope with very large networks and continuous monitoring of applications [46,47].

There are some research works that survey the state-of-the-art in the field of scalable networks for heterogeneous systems, software-based networking, and hybrid systems involving several categories of smart devices, such as [48], where the authors present studies of ML/DL applications in software-defined environments.

The methods for assessing the network performance may be split into two categories:-active methods for network efficiency and level of service monitoring, involving the injection of probe traffic into the network to learn about its state of operation, as well as-passive methods, observing and analyzing different KPIs collected in big data storages.

Table 1 presents in a comparative mode some of these aspects.

Taking into consideration the information presented in Table 1, it is obvious that a combination of the two techniques is the most beneficial for the NTMA. However, this is a complicated process to implement because it needs a deep understanding of the network and messages structures, and for this to become effective, a very complex team of experts with a period of accommodation, or training, is also needed. 

Complex systems and distributed network maintenance have also been the preoccupations of many researchers [48,49,50], and modeling of the present and future states using different models, including Markov Chains and/or Hidden Markov, are discussed in connection with some applications for several systems [51], based on the modeling of hidden states of those systems. These solutions might involve complex algorithms and also presume higher computation power for achieving usable results in the prognosis of a system’s future states, as well as possible training, using simulated or collected data. Markov Chains and Hidden Markov Chains (HMMs) are both mathematical models used to describe stochastic processes, where the state of a system evolves over time. The Markov Chain consists of a finite set of states and a transition probability matrix. The matrix defines the probability of transitioning from one state to another. Each state has a fixed set of transition probabilities associated with it, and these probabilities remain constant throughout the process.

A Hidden Markov Chain is an extension of the Markov Chain model that incorporates hidden or unobservable states. In a HMM, the system has a set of observable states, but the underlying state of the system is hidden or unknown. The observed states are generated by the hidden states through a set of probability distributions. In general, HMMs require more processing power than simple Markov Chains due to the additional complexity involved in inferring the hidden states from the observed states. The computational complexity of HMMs arises from the need to estimate or infer the hidden states using algorithms, such as the Viterbi algorithm or the Baum-Welch algorithm. Moreover, HMMs often involve more complex probability distributions for emission and transition probabilities compared to the constant probabilities in simple Markov Chains. These probability distributions usually require additional calculations and more processing power to handle.

The present work is focused on proposing an integrated platform for preventive maintenance, which is dedicated to complex smart city services and involved data communication networks, based on a less demanding computation power. Therefore, it uses only observable indicators, using data collected by different intelligent agents. These agents harvest information both from hardware and communication channels loads, as well as from the applications’ availability and response times. As a continuation of a previous research [52], the use of intelligent agents in early discovering and noticing deviations of normal operation and lowering of the level of service is associated in this work with the updating of a current state matrix and computing different state probabilities for a future state prediction matrix. The latter is aimed at providing the operator with alerts and suggestions for alleviating malfunctions’ and maloperations’ negative effects.

The remainder of this article is organized as follows: Section 3, Materials and Methods, describes the main aspects regarding the permanent monitoring of reliability and levels of service based on Markov Chain modeling of a future state matrix. Section 4 proposes an algorithm for integrating the state matrix and clients’ satisfaction in a common monitoring platform, as well as application on a case study with six smart city services, and, finally, Section 5 and Section 6 are proposed, where an analysis on the utility of the proposed solution is discussed, along with future developments.

## 3. Materials and Methods

### 3.1. Reliability and Maintenance Relationship

Due to their required high level of service, smart city services and supporting data communications networks need permanent monitoring and maintenance. Due to the continuous development and the growing complexity, these networks have become difficult to monitor and maintain.

Therefore, there is a need for automated maintenance processes, supported by intelligent agents able to early detect failures, malfunctions, and any other defective operations. At the same time, even manual upgrading, deployment of new software versions, operational support, troubleshooting, etc., may become sources of defective operation of some of the functional components from the complex networks. In fact, as personal observations reveal, on some of the mobile communication networks in Romania, intensive upgrading and improvements in the functional (hardware or software) components caused more than 55% of the events causing low levels of service. This might be somehow justified, considering the vast complexity of the network and implications that one server, or application, have in the overall process, implications that the human personnel might not be able to envisage from the beginning. Moreover, there are some causes that cannot be forecasted, such as natural disasters (flooding, earthquakes, fire, etc.), or works in the field, performed by other parties, which are possible to intrude in the physical cabling, but these seem to be much rare than the functional failures or human intervention effects, such as third-party vendor services failures, security breaches, and so on.

For achieving the goal of obtaining a simple solution for the integrated management of hardware failures, software problems and customer satisfaction, in this research, the following aspects have been addressed:-Integration of (existing) intelligent agents for hardware, applications, and services monitoring-Proposing an algorithm for building a state matrix for the system-Proposing an algorithm for building and updating the state-transition matrix, based on the Markov approach-Development and adaptation of the solution for the clients’ satisfaction analysis-Integration of all these approaches in a single platform to assist the maintenance operators in early detection of malfunctions and network decrease in performance-Creating a risk evaluation matrix for the maintenance operations

In general, the probability of failure is best described in reliability theory by the failure rate,
(1)$$\lambda \left(t\right)=\sum_{i=1}^{N}\lambda_{i}\left(t\right)$$
where; $\lambda_{i}\left(t\right)$ represents the failure rate of the independent functional component, and N is the total number of functional components taken into consideration. We say that $\lambda \left(t\right)$ is a probability that the product will work without failure until the considered moment and fail during the immediately following time unit (if this unit is small).

Then, the overall reliability function $R_{T}\left(t\right)$ of the system for a year is given by:(2)$$R_{T}\left(t\right)=e^{-\sum_{i=1}^{N}\lambda_{i}\left(t\right)\cdot t}$$
where; *t* is the duration of time corresponding to a year, expressed in hours, and the mean time between failures (MTBF) is given by:(3)$$MTBF=\frac{1}{\sum_{i=1}^{N}\lambda_{i}\left(t\right)}$$
(if the chain of reliability only considers the equipment) Because determining the utility function (failure-free operation) requires a large volume of experience, the reliability of a product is generally characterized by the average duration of operation:(4)$$T_{0}=M\tau =\int_{0}^{\infty }tq\left(t\right)dt=-tP\left(t\right){|}_{0}^{\infty }+\int_{0}^{\infty }P\left(t\right)dt=\int_{0}^{\infty }P\left(t\right)dt$$
where; $M\left[T_{d}\right]$ or $M\left[T_{i}\right]$ represent the average value of a repair or replacement of time between two consecutive successful states of operation, during which the respective installation repaired or replaced, and $P\left(t\right)$ represents the probability that the product will work without breaking down until time *t*: $P\left(t\right)=P\left(\tau \geq t\right)$.

For example, in a mobile communication network in Romania, there have been cases where applications which were making requests to a specific public domain failed because other, banned domains, stole the public IP of the legal one, a process which led to blacklisting the correct IP. Intensive maintenance of complex networks, consequently, could also produce negative effects, such as randomly lowering of some service levels, increasing operation costs, causing outage duration costs, etc. A balance is necessary to be made between maintenance costs and outage duration costs.

The most appropriate maintenance service can be determined via two different approaches:-Preventive maintenance—via scheduled procedures, condition-based procedures, or reliability-centered maintenance-Corrective maintenance—operation is performed after the failure has manifested. It might also trigger corrective measures, or changes in the structure of the network, upgrading of software components, etc.

Mathematical modeling of maintenance should consider an objective function, seeking an optimum between the following criteria: minimization of restoring time, minimization of maintenance costs, and risk minimization. It is considered that a model which employs risk management is important in AI-assisted preventive maintenance, being more efficient in suggesting the human operators the appropriate measures to be taken and their forecasted risks in terms of operating levels of service for the different hardware and software components. This is because the quantification of risks enables determining an optimal level of risk, which provides the most efficient maintenance strategy for complex systems and networks.

The methodology in this paper proposes the automation of multiple integrated processes, namely, (i) the introduction of risk assessment-based functional monitoring agents and (ii) the monitoring of the clients’ satisfaction. To determine an optimal preventive maintenance objective, it is necessary to analyze multiple possible operating states and scenarios, based on state transition matrixes. A multi-level approach is easier to introduce in practice, especially when complex networks and services are involved. In this way, a dedicated monitoring application and model should be developed for the data communication network. Then, a higher-level application for monitoring complex services (including the monitored network) is to be set on a superior level of implementation. This superior-level application shall be in charge, also, of monitoring clients’ satisfaction.

### 3.2. Building the Algorithm for Network and Service Risk Assessment

This subsection describes the proposed approach for obtaining an automated preventive maintenance process, helping human operators in the fast recovery activities of the data communication network, or preventing the occurrence of a failure, due to early warning messaging. 

The basis of this model is founded on the analysis of a complex data communication network and a set of relevant smart city-related monitoring agents, and, from the point of view of the operating states, the main causes of the decrease in the level of some services, as well as the analysis of the causes of the most frequent hardware and/or application failures, are considered. A transition matrix is then built, considering different failure rates and the corresponding risk factors, with associated causes. Risk is defined as the product between the probability that a failure occurs and the expected value of costs that the failure produces in the system. The risk is defined at the level of the considered data network. The evaluated data network is a complex one, with different services and applications, and it is used as a backbone data communication network in a smart-city environment, where different services also rely on smaller communication networks, such as ZigBee, Bluetooth, or LoRa.

For the intelligent monitoring of the backbone data network, previous work results have been presented in [52]. The following intelligent agents have been in use for monitoring smart city services:Traffic service levels monitoring serviceEnergy distribution service levels monitoringEnvironment monitoring serviceCrowdsourcing monitoring servicePublic lighting monitoring serviceWaste disposal monitoring service

Each individual agent is set to monitor a specific service from the point of view of its functionality, iteratively, and/or by event triggered. Each record is indexed with event start and event end timestamps to determine the service unavailability duration. The assessment took place for a one-year period, during which all six services have been monitored from the availability point of view, namely, the ratio between the count of successful requests of the service and overall requests (successful plus failed requests to services). The diagram presented in Figure 1 shows a sample analysis for a month period, where a specific service has experienced some failures. The number of failures is represented by the vertical (blue) bars, while the availability index is presented in the upper part of the diagram, in percents, and the red line shows the evolution in time of these indexes.

The next figures present, in detail, samples of the six independent service activities during the monitoring period: Figure 2—traffic monitoring service, Figure 3—energy distribution monitoring service, Figure 4—environmental monitoring service, Figure 5—crowdsourcing monitoring service, Figure 6—public lighting monitoring service—vertical lines represent the division of time for monitoring the service due to the fact that this specific service is only monitored during night time, and, finally, Figure 7—waste disposal monitoring service. The red vertical lines represent decreases in services’ availability due to the different causes, including malfunctions, equipment failures, maintenance operations, software upgrading, OSI physical level degradation, etc. 

Some of the most common failures noticed have been caused by human interventions, including corrective maintenance, curative maintenance, software upgrading, preventive maintenance, peer migration, hardware replacement, hardware upgrade, and standardization.

Considering the impact of these malfunctions, the following represent the main effects, on a scale from the worst to the less harmful impact: complete failure, traffic loss, incoherence/loss of data, latency, loss of administration, loss of supervision, mini failure (complete failure for a max. 10 min time), and slow response.

The probability of uninterrupted functioning for this service, computed based on collected data, was $P_{tm}=0.99725$.

The probability of uninterrupted functioning for this service computed, based on collected data, was $P_{ed}=0.997407$.

The probability of uninterrupted functioning for this service computed, based on collected data, was $P_{env}=0.99805$.

The probability of uninterrupted functioning for this service computed, based on collected data, was $P_{cs}=0.999448$.

The probability of uninterrupted functioning for this service computed, based on collected data, was $P_{lm}=0.996346$.

The probability of uninterrupted functioning for this service computed, based on collected data, was $P_{wd}=0.998662$.

For the whole set of services, the overall probability of uninterrupted functioning reached the value of $P_{S}=0.987227$. The most affected months by services’ dropdowns were August (four warnings, with service availability of less than 99.5%), October (two warnings) and December (two warnings). Causes of these reductions in service availability might include: promotional campaigns and deployments, creating collateral problems, slower response to failures due to lack of personnel (August), weather conditions (December), insufficiently documented and bad organized preventive, and /or scheduled maintenance operations (all cases).

The proposed approach is based on analyzing the processes with Markov Chains, for the states in which the (super-) network (i.e., network of networks) could evolve, based on events tracked over a determined period. The established quantized states in which the (super-)network could evolve are the following: 100% functional (no failures), degraded level 1 (small service degradations, acceptable—e.g., delay in service delivery), degraded level 2 (missing some non-essential services), degraded level 3 (missing some essential services), and fully degraded (no service).

The approach was developed in two directions:(i)Reliability analysis(ii)Client satisfaction analysis

In practice, the algorithm for reliability analysis works based on the following processes:-Process 1: extracting information regarding the availability of the services on the determined period, to observe eventual patterns, and creating a table with agents’ availabilities, also containing the average outage probabilities-Process 2: detecting the transition from the current state to another state and creating a database table with these transitions-Process 3: calculation of the matrix of state transitions based on Markov Chains-Process 4: executing subroutine for defining the risk levels depending on the transition probabilities between the states
$$\begin{array}{cc} \mathrm{Case P} & (\mathrm{Current}\;\mathrm{state}=>\mathrm{Possible}\;\mathrm{state}\;x)\;\mathrm{between}\;\{\mathrm{interval}\;1\},\mathrm{Risk}\;\mathrm{level}=“\mathrm{Very}\;\mathrm{Low}”\\ & (\mathrm{Current}\;\mathrm{state}=>\mathrm{Possible}\;\mathrm{state}\;x)\;\mathrm{between}\;\{\mathrm{interval}\;2\},\mathrm{Risk}\;\mathrm{level}=“\mathrm{Low}”\\ & (\mathrm{Current}\;\mathrm{state}=>\mathrm{Possible}\;\mathrm{state}\;x)\;\mathrm{between}\;\{\mathrm{interval}\;3\},\mathrm{Risk}\;\mathrm{level}=“\mathrm{Medium}”\\ & (\mathrm{Current}\;\mathrm{state}=>\mathrm{Possible}\;\mathrm{state}\;x)\;\mathrm{between}\;\{\mathrm{interval}\;4\},\mathrm{Risk}\;\mathrm{level}=“\mathrm{High}”\\ & (\mathrm{Current}\;\mathrm{state}=>\mathrm{Possible}\;\mathrm{state}\;x)\;\mathrm{between}\;\{\mathrm{interval}\;5\},\mathrm{Risk}\;\mathrm{level}=“\mathrm{Very}\;\mathrm{High}” \end{array}$$

For the present case study concerning the reliability analysis, to obtain information regarding the probabilities of transition between these states (transition matrix), a period of one year of analysis has been assessed, with a sampling interval of one minute. The data has been collected from a large network and services operator. For each sample, the current state (according to the six possible) has been recorded, along with the timestamps. The developed algorithm (Figure 8) extracted information regarding the types of transitions (from previous state to the new one), and, with the results, the transition matrix has been built for the analyzed period.

The algorithm for the current state assessment (Figure 8 upper part) develops as follows:-Intelligent agents collect information on current states of the services and network—either on specific moments (regularly reading), or by event-triggered.-The state matrix is built and updated constantly, based on recording the state transitions: from the former state in the new, current state, marking each state transition with a flag in the matrix (Figure 8, lower part indicates an example of transitions). The corresponding cell of the matrix (where the line index represents the former state number, and the column index represents the new state number) will be incremented.-On a repetitive basis, the number of transitions between different operational states are computed and transformed into transition probabilities. In time, the state matrix improves in estimating the probabilities of transitions.

The algorithm for the client satisfaction assessment works similarly, with the following differences:-The evaluation criterion for establishing state transition is in this case the *APDX* index, given by Dynatrace application-The following thresholds were established:
○75%<APDX≤80%—*S_n_*_4_—Catastrophic state.○81%<APDX≤88%—*S_n_*_3_—Severe degradation state.○89%<APDX≤94%—*S_n_*_2_—Degradation state.○95%<APDX≤97%—*S_n_*_1_—Graceful degradation state.○98%<APDX≤100%—*S_n_*_0_—Normal operational state.-The state matrix for the clients’ satisfaction is built and updated constantly, based on recording the state transitions: from the former state in the new, current state.-The next transition (e.g., from a defective state into the fully operational state) is also marked as the new state.-On a repetitive basis, the number of transitions between different operational states are computed and transformed into transition probabilities. In time, the state matrix improves in estimating the probabilities of transitions.-Based on the recorded transitions in the transition matrixes, the inherent and residual risks are evaluated and displayed to the operators, showing a risk rating ranging from “Sustainable” to “Critical”.

The following, Table 2, shows the format for the transition matrix, where $pn_{xy}$ represents the probability of changing from state *x* into state *y*.

In Table 2, SNx represents each possible state, according to the definitions previously given. For example, SN_1_ might represent that a local network has a longer response time, SN_2_—host domain for several services from main network are down, SN_3_—physical hardware in data center is malfunctioning, or multiple IP addresses are inaccessible, or routing rules are working improperly, and SN_4_—physical level damage of the OSI stack occurred, or there is a huge increase in all requests from clients without response. Based on the data collected from the case study, the following results have been achieved for the availability of services (Table 3, values in percents). 

To give a more comprehensive image of the processes’ availability, the numbers in Table 3 have been transposed in diagrams, where the blue columns represent the total minutes of count in the respective month, and orange columns represent service up-time recorded minutes during a month (Figure 9, Figure 10 and Figure 11). 

The numerical data presented in the above table was obtained by monitoring, for a period of one year, the most important services in a communications network and recording all types of incidents that led to their degradation. The amount of degradation suffered by services (representing the low value of availability) was analyzed, as well as the number of incidents and their duration. Each transition of the services from a working state of 100% availability to any other state of degradation was counted on each individual type, as well as the transitions from intermediate states of high degradation to those in which services are almost recovered. In all this analysis, the duration of each incident and its impact are important.

Incidents/fails are most often detected by applying APM methodologies (Application Performance Management) with agent-based and AI monitoring tools, such as Dynatrace. Without these tools, technical teams often find it difficult to find the root cause of an application performance problem.

However, sometimes, some incidents are detected reactively by being informed by third parties of the existence of a problem or by observing the increase in the number of complaints. The operational teams that deal with maintaining the availability of applications as high as possible, 24/7, use a bunch of monitoring and alerting tools for information and quick intervention in the event of an incident. For each incident, the moment of beginning, its severity, the impact on services, and the moment of recovery, are recorded. After full recovery is achieved, most of the time, through reverse engineering or analysis of the last interventions made on the system, the cause of the production is also analyzed and noted.

The values in Table 4 were collected based on the historical database querying and represent the probabilities of transitioning between a specific state (of degradation) into another state, representing the number of events recorded during the one-year period of analysis.

Based on the current state of operation acquisition, and the state transition matrix, it becomes now possible to estimate the future state of the network after *n* sampling steps in the future, using the Markov Chains approach: $S_{n}=p^{n}S_{0}$, where $S_{n}$ is the probability of the predicted state at the *n*th sampling moment, $p^{n}$ is the transition matrix raised to the power *n*, and $S_{0}$ is the probability of the current state. The final goal is to create a risk assessment mechanism for improving the preventive maintenance process. As an example, using the collected data, the future predicted state after two sampling periods $(S_{2})\;$is presented in Table 5.

### 3.3. Buliding the Algorithm for Clients’ Satisfaction Forecasting

Using the Apdex scores (*APDX*), the satisfaction of clients regarding the different services has also been assessed for a year period. *APDX* are defined by the ratio between the sum of satisfactory and tolerated requests over the total requests made in the analyzed period (one year, monthly averaged):(5)$$APDX=\frac{S_{R}+0.5\cdot T_{R}+0\cdot U_{R}}{N_{R}}$$
where; $S_{R}$ stands for the number of satisfactory requests, $T_{R}$—the number of tolerable requests, and $U_{R}$—the total number of unsatisfactory requests.
-Satisfied—satisfied client having a high application responsiveness. (depending on application, less than 1 s, typically tens of milliseconds)-Tolerating—a client with noticeable slow response from the application (depending on application, less than 5 s, typically in the range of 1–3 s)-Unsatisfied (frustrated)—a client experiencing unacceptable performance, leading to abandonment of the application (typically more than 5 s)
(6)$$N_{R}=S_{R}+T_{R}+U_{R}$$

However, it is important to keep in mind that different categories of clients might have different expectations for the services or applications’ performance. It is crucial to create useful scores for the experiences the clients would expect, and this is mostly a human operator-based experience. The clients will be willing to wait if the service brings something desirable at the other end while in other areas, where they are not enjoying the process, maintaining a high *APDX* score might prove crucial.

For the same analysis period, the *APDX* index has been calculated for all six services mentioned above. The results are presented in the following table.

To give a more comprehensive image of the clients’ satisfaction regarding the services, the numbers in Table 6 have been transposed in diagrams, where the columns represent the measured levels of satisfaction (Figure 12, Figure 13 and Figure 14). The dotted line represents the trend over the entire analysis period (averaged trend over one year), in all figures below.

The above, Figure 12, Figure 13 and Figure 14, help turn measurements into insights on how satisfied the clients are in a smart-city environment of e-services. It is an addition to the quality of service in the overall preventive maintenance process, especially when upgrading different software components to offer the best satisfaction to clients. Further analyzing the trends over the test period, one can see what services need attention and possible upgrading.

## 4. Results

### An Algorithm for Building the Risk Assessment Matrix

The purpose of this approach was to design an automated solution for data collection regarding the state of operation of several networks and services (collected from AI agents) to help maintenance operators in decision making, based on future possible risks prediction. In this section, the building of an AI-driven risk assessment matrix is presented, and a global information table containing risk ratings, with or without any control measures, is presented. Additional information may include responsible departments and recommended actions. In this approach, residual risks are considered those that might still occur after the first set of maintenance operations has been performed.

Based on the above obtained results, an AI-driven risk assessment matrix has been designed, via the following processes, presented in Figure 15. The functional block, representing a Markov process (computation of transition state matrix) has been presented in more detail in Figure 8 (previously), showing that each state change in the network is counted, the new state is recorded at regular time intervals, and the probabilities of state changes are re-computed to update the whole matrix.

Building a table to show the mapping of impact degrees (on 5 levels—very low, low, medium, high, very high) and the related probabilities of occurrence is presented in Table 7.

In Table 7, the associated colors are intended to help the operator to rapidly assess the critical situations, seeing the gravity of an event without reading the risk.For each of the services and networks’ present states, $S_{N}$, a table with possible transitions to next states, $S_{N+1}$, and their associated risks, is constructed.Computing the residual risk levels (the risk of passing into a non-functional state, partially or totally, following the restoration interventions already applied)Displaying residual risksDisplaying current operating status: if the current state is not 100% operational, then display actions, recommendations, alarms, and involved departments. Then, display the most probable next state, according to the forecast.Display, gradually decreasing, the pre-calculated risk levels for transitions in all other possible states.

Taking into consideration previous experience in maintenance works, the algorithm has been enhanced with a supplementary risk analysis feature, that is, the Residual Risk with Control (RRC) matrix. This feature should improve the vision of the maintenance staff with more actions to avoid collapsing into a new failure situation of the network and/or services, when inappropriate actions during the recovery might trigger cascading failures, or disturbing other services that were previously in a good functioning state. To obtain less faulty results in recovery actions, recordings of failures caused by incorrect, or insufficiently documented maintenance operations, could be used to build a probability matrix. Based on this database, a model for evolving states of the system might be obtained via a similar process to the previously described one.

Any evolution in time and changes in the current state of the system is being recorded according to the policy adopted for this protocol: regular sampling of states and/or event-triggered recording of state. Due to this policy, temporal features may be also part of this recorded information and further analyzed to discover if the system might experience periodic temporal patterns that respond to a specific behavior. This process could help maintenance service to perform upgrading and/or corrections in the system to avoid such behavior.

For the case study in analysis, Figure 16 presents a screenshot with the AI-driven risk assessment matrix, completed with the section of risk assessment with control. Additionally, this matrix can be considered a useful instrument in preventive maintenance, serving as a tutorial for:building new regulations for assessing the risks to which subsystems, or services might be affected when periodical maintenance interventions are performed,defining new operation procedures,creating standardization, etc.

In Figure 16, the following rates of risk have been considered:

Very Low—risk rate $R_{r}\leq 20\%$.Low—risk rate $21\%\leq R_{r}\leq 40\%$.Medium—risk rate $41\%\leq R_{r}\leq 60\%$.High—risk rate $61\%\leq R_{r}\leq 80\%$.Very High—risk rate $81\%\leq R_{r}\leq 100\%$.

Each system state change is detected automatically (by the sudden change in the values of the availability and APDX indexes) and, after such a change, a new line with the current state is recorded in the matrix (current state ID). For this new state, the possible future states are calculated, and these include the probability of passing into them, the impact that passing would have, and, based on these last two criteria, the estimated risk is displayed if no corrective measures are being taken. Retrieving information from adjacent monitoring tools, such as Dynatrace, can also help detect the possible main causes that led to the change in condition.

Based on the history related to the impact that the application of the basic corrective measures had in the past, the possible future states with control and the risk of their occurrence are computed and displayed.

In both cases, the risk assessment tool offers the user an overview of what the evolution of the system/network could be and informs him/her which are the departments responsible for corrective measures. The AI-DRAM shows a general view of how many low risks, medium risks, or very high risks the maintenance operator could have to be able to cope with different situations and choose the most appropriate measures, according to the moment of maintenance. This tool immediately gives the whole table of the possible implications that the maintenance operation could involve. According to the case study mentioned in this work, Table 8 shows the occurrence of events, which produced failures (colour in background meaning an information on the gravity resulting from the combination between risk probability and its impact: green—very low, yellow background—low, brown background—medium, and red background high/very high).

In Table 8 is depicted the table that quantitatively describes the historical evolution of the events which produced failures, based on the correlation between the impact and the probability of occurrence.

Since these still represents early research results, which are based both on real world and simulated data, it is difficult, at this moment, to present, in a comparative mode, how the proposed solution gives better results than similar, existing methodologies. However, according to the authors’ actual knowledge, similar solutions only focus on solving specific problems, such as those of network level of service, application, and/or clients’ satisfaction monitoring in a separate, and not in an integrated manner. This study started from the desire to find a solution to the numerous functionality problems mainly caused by human triggered actions, updating of services, and re-allocations of resources, which led to chain effects in reducing the network level of service and clients’ satisfaction. The approach focused mainly in seeking for a simple way to deliver an integrated solution, without resorting to complex AI algorithms, but keeping a desired level of support for assisting the failure management system, based on a passive approach and data mining. One of the main reasons for choosing this approach was to reduce the complexity of the software programming, and to also provide a reliable strategy for data updating and post-processing analysis, thus reducing as much as possible the involved resources. The proposed algorithm is only updating the state matrix at each iteration of the process for diminishing the computational overhead.

## 5. Discussion

Complex network maintenance modeling is presently an active area of research. Various approaches and techniques are being explored to address the challenges associated with maintaining complex networks. In this field, the main directions of research include:-Reliability-based maintenance modeling—by the quantification of the reliability and availability of complex networks and optimization of the maintenance strategies accordingly-Prognostics and Health Management (PHM)—oriented towards the prediction and prevention of failures in complex networks by continuously monitoring the health condition of network components-Condition-Based Maintenance (CBM)—strategies that rely on real-time condition monitoring and diagnostics to optimize maintenance decisions-Stochastic modeling and simulation-Optimization-based approach-AI and data-driven approaches, etc.

Usually, a combination of these approaches might prove more effective in finding optimal solutions for complex network maintenance modeling.

The present work represents a continuation of research [51], as well as a combination of PHM-CBM, focusing on the development of an automated tool to assist maintenance operations for complex, heterogeneous systems, and data communication networks. Regarding the complexity of automation components, IoT-connected devices and communication networks increase on a day-by-day basis, and the maintenance of heterogeneous systems becomes more and more difficult. Therefore, the support that an automated maintenance process could bring is considered beneficial in increasing the productivity and resilience of IoT-based smart city services. In this research, a solution for assessing the risks and estimating the future states of the complex environment of a smart city has been developed. When intelligent monitoring agents for hardware and software components and users’ satisfaction are employed, harmonization of recorded events regarding malfunctions and maloperations should be subject to automated processing, too. For this to be achieved in an efficient manner, the present research developed a risk assessment matrix, based on a one-year analysis of a case study comprising six smart city services and related data communication networks. The purpose of this analysis was to create evidence of the services’ availability and of the APDEX.

After this analysis has been completed, a second quantitative assessment of the causes of incidents and how these incidents affected the functionality of the system has been made. Any incident that occurred caused a decrease in the performance indicators. However, a special interest has been the analysis of the causes that produced incidents. It has been emphasized that, in this study, human interventions (maintenance operations) that caused incidents have been considered.

The next phase was to count the number of incident occurrences and to compute their probabilities. Based on the obtained probabilities, a state transition matrix has been developed. Using a Markov Chains approach, future possible states of the system could be then estimated. For each of the malfunction states, an impact has been associated, and a scaling of the respective occurrence probabilities has been proposed (as presented in Figure 16). The AI-DRAM has been developed, based on a dedicated algorithm. In this algorithm, anytime the current state is changing, a new record is registered, and the new, estimated probabilities of the system to evolve in any possible state is computed, along with the associated impacts. The main KPI of this process is the availability indicator; however, the APDEX is an improvement to correlate services quality with the clients’ satisfaction. Compared with similar technologies, the present solution has some advantages (Table 9).

The proposed algorithm is not considered a perfect solution, and it is very possible that similar, modern research solutions may surpass it in efficiency in some respects [52,53,54], such as employment of Adaptive Transmission Data Rate (ADTR) mechanism or Self-Adaptive Routing Algorithm (SARA) or the use of a Partially Observable Markov Decision Process (POMDP) formulation to map metrics to 5G requirements, aimed to improve reliability. However, similar work [55] also confirms the applicability and efficiency of Markov Chain State Prediction, compared with other algorithms, such as sequential Monte Carlo, in estimating even k-out-of-n systems’ reliability, based mostly on its reduced computation time. 

Some of the main advantages of the solution given by Markov Chains prediction, which stood at the fundament of taking this approach, are the following:-Capturing State Transition Probabilities: by analyzing historical data, one can extract state transition probabilities and build a Markov Chain model that accurately represents the network’s dynamics. This allows one to make informed predictions about future network states with high precision.-Simplicity and Computational Efficiency: this property simplifies the prediction process, becoming unnecessary to consider the entire history of the system. Additionally, Markov Chain models are computationally efficient, enabling real-time or near-real-time predictions, which are crucial for dynamic communication networks.-Flexibility and Adaptability: scalability is a significant advantage of using the Markov Chain for future state prediction in communication networks. These networks often comprise a vast number of interconnected elements, such as routers, switches, and transmission links. Markov Chain models can handle large-scale networks without compromising prediction accuracy. By dividing the network into smaller manageable subsystems, we can build local Markov Chain models and aggregate their predictions to obtain an overall network forecast. This approach ensures scalability while maintaining accuracy.-Decision Support and Optimization: predicting future states in communication networks involves making informed decisions to optimize network performance. Markov Chain models can serve as decision support tools.

In conclusion, the main benefits of this research include, compared with similar work: integrating a solution for both systems state and applications response time monitoring, analysis and evolution, and employing a less computationally intensive approach.

Whether Hidden Markov Models (HMMs) are more accurate than simple Markov Chain models depends on the specific problem and the nature of the data being modeled. In some cases, HMMs can provide more accurate results, while, in others, simple Markov Chain models may be sufficient or even preferable. For this specific case, the simpler solution has been preferred with regards to the idea of reducing the complexity of the processes and required computational stress. Simple Markov Chain models were considered more appropriate for this case because the problem involves directly observable states and the transitions between those states. This is the case when modeling the behavior of a system with a fixed set of states, where the future state was considered to depend only on the current state and may not require the complexity of HMMs. However, it is possible that, based on the future experience, the choice between HMMs and simple Markov Chains for developing this approach should be based on a careful analysis of the problem domain and the specific requirements of the application.

The main limitations of the proposed solution consist in the difficulty of initializing the state matrix because this is a process which needs previously collected and sorted data on hardware and applications failures. The clients’ satisfaction section, however, can be based on directly monitoring of different metrics and using specific tools, such as Dynatrace, and related KPIs, such as APDX. However, in a correct implementation, it is estimated that the solution can improve over time, when sufficient data is collected, allowing for a better analysis of the system’s behavior.

It is desirable that, in the future, the standardization of these solutions becomes more practical and mandatory, as the heterogeneity of AI-assisted maintenance of complex networks and system will probably become too large to harmonize intelligent systems and the exchange in information. However, some trends in this regard have been noticed, such as the Matter standardized application layer for connections in the IoT.

## 6. Conclusions

It is believed that the optimal combination between hardware sensors and intelligent agents can reach the highest degree of dependability in the complex process of preventive maintenance. However, it is not easy to implement active AI-driven solutions due to intensive initial resources requirements: staff training, specialists’ involvement, complex teams building, good knowledge of system architecture and behavior, etc. A simpler start is proposed in this research by using passive maintenance monitoring. The approach is based on historical data analysis and artificial intelligence (AI) to anticipate and prevent failures, reduce downtime, and optimize performance. In this research, the goal was to try to optimize the interaction between automated and human-performed operations maintenance.

Presently, the solution has been applied on a set of six smart city related services, collecting and analyzing information recorded in one year. Only for the communication network services, considering the obtained results based on the maintenance staff feedback, it can be concluded that it helped in improving the recovery time of major failures with a percentage of 9.4% in 87% of analyzed cases, including human interventions and network scheduled maintenance procedures.

Future research will focus also on providing automatic network configuration solutions to maintain the level of service within acceptable limits. 

## Figures and Tables

**Figure 1 sensors-23-06012-f001:**
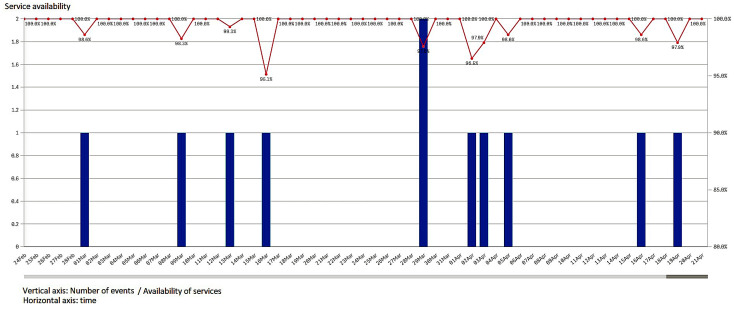
A sample of availability monitoring chart during a period of two months.

**Figure 2 sensors-23-06012-f002:**
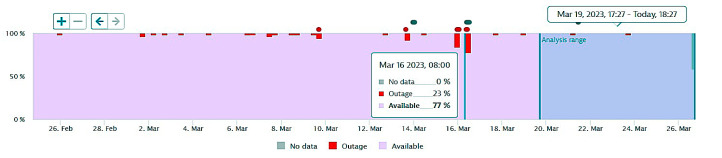
The traffic monitoring service availability on the monitored period (sample).

**Figure 3 sensors-23-06012-f003:**
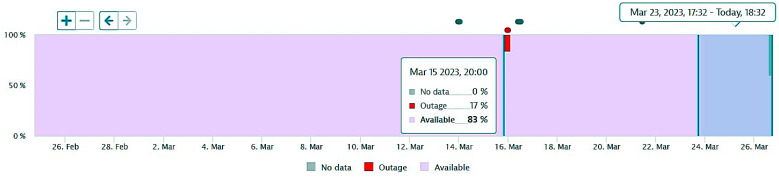
The energy distribution service availability on the monitored period (sample).

**Figure 4 sensors-23-06012-f004:**
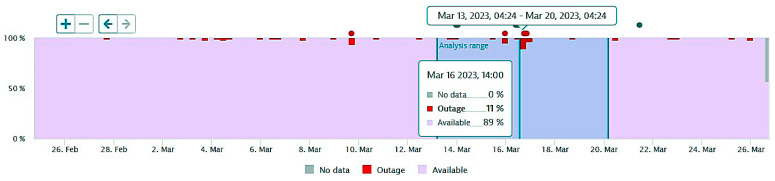
The environmental monitoring service on the monitored period (sample).

**Figure 5 sensors-23-06012-f005:**
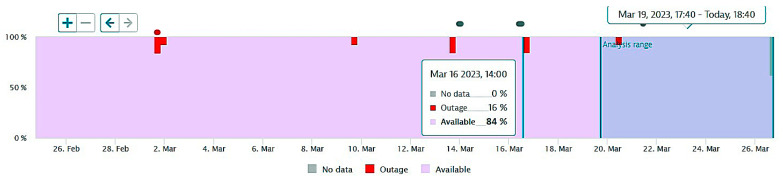
The crowdsourcing monitoring service on the monitored period (sample).

**Figure 6 sensors-23-06012-f006:**
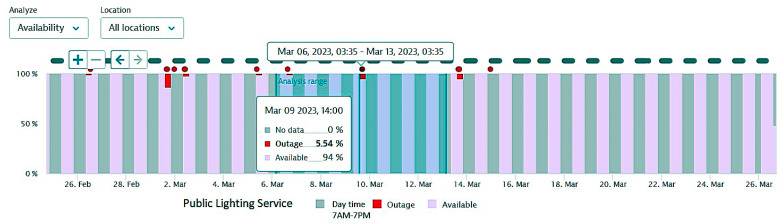
The public lighting monitoring service availability (nighttime only).

**Figure 7 sensors-23-06012-f007:**
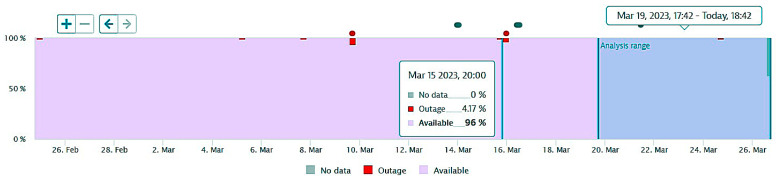
The waste disposal service availability on the monitored period (sample).

**Figure 8 sensors-23-06012-f008:**
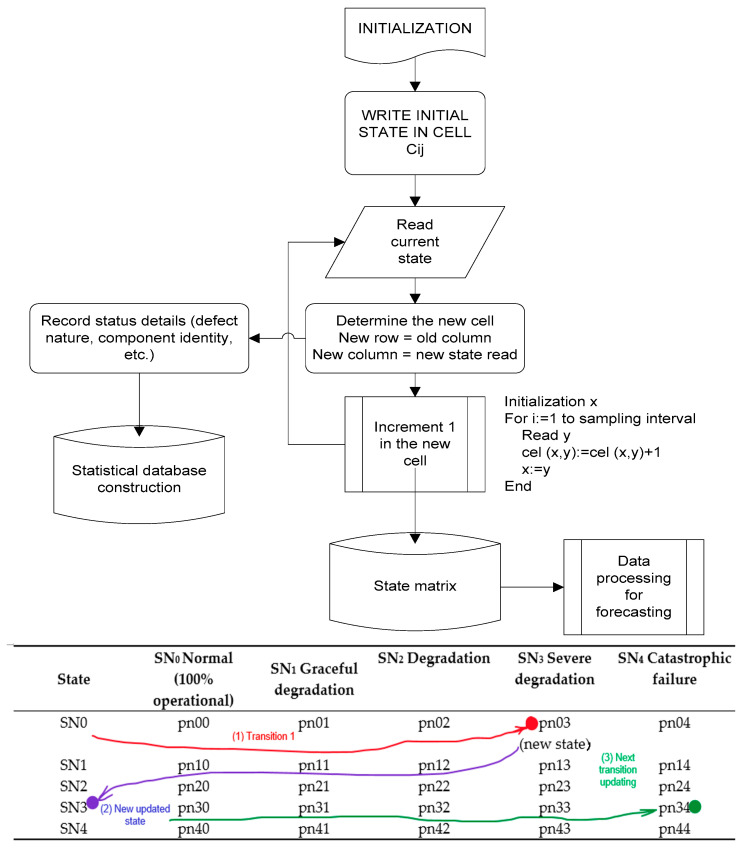
The algorithm for building the transition matrix.

**Figure 9 sensors-23-06012-f009:**
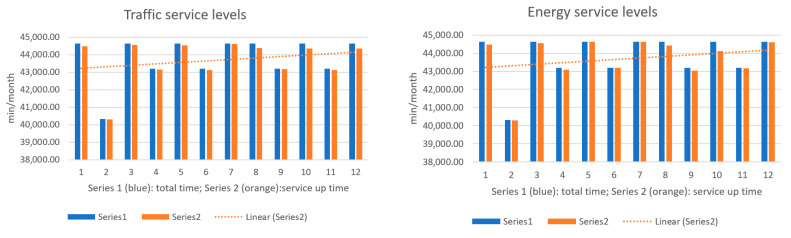
**Left**: traffic service availability on the test period; **Right**: energy service availability on the test period.

**Figure 10 sensors-23-06012-f010:**
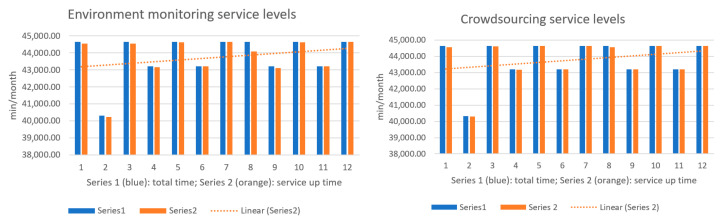
**Left**: environment monitoring service availability on the test period; **Right**: crowdsourcing service availability on the test period.

**Figure 11 sensors-23-06012-f011:**
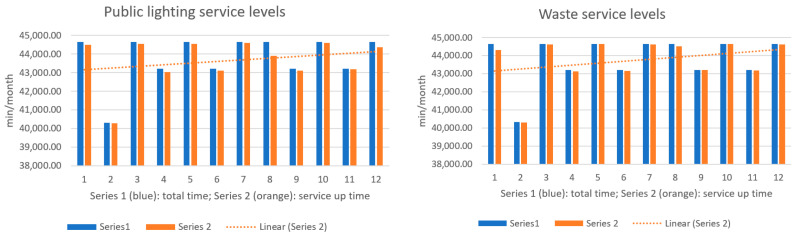
**Left**: public lighting service availability during the test period; **Right**: waste service availability during the test period.

**Figure 12 sensors-23-06012-f012:**
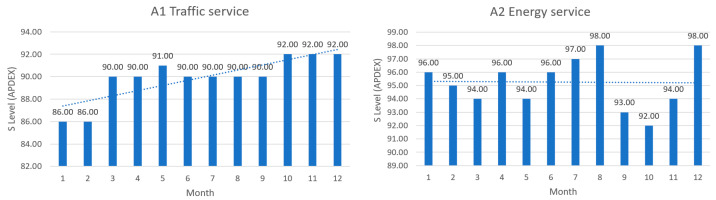
**Left**: traffic service clients’ satisfaction in the test period; **Right**: energy service clients’ satisfaction in the test period.

**Figure 13 sensors-23-06012-f013:**
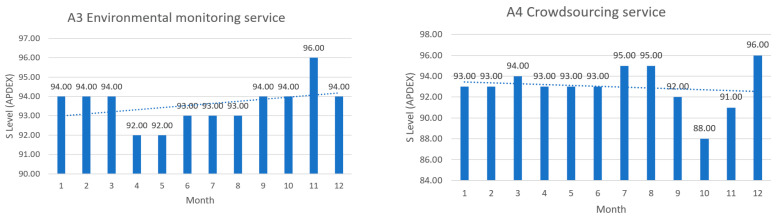
**Left**: environmental monitoring service clients’ satisfaction in the test period; **Right**: crowdsourcing service clients’ satisfaction in the test period.

**Figure 14 sensors-23-06012-f014:**
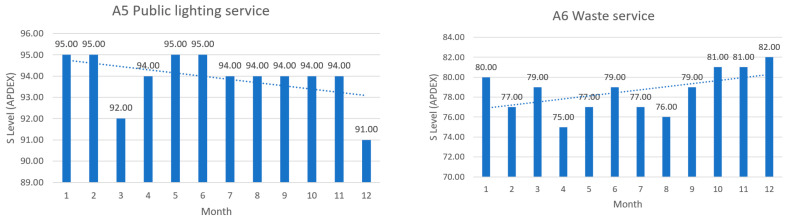
**Left**: public lighting monitoring service clients’ satisfaction in the test period; **Right**: waste management service clients’ satisfaction in the test period.

**Figure 15 sensors-23-06012-f015:**
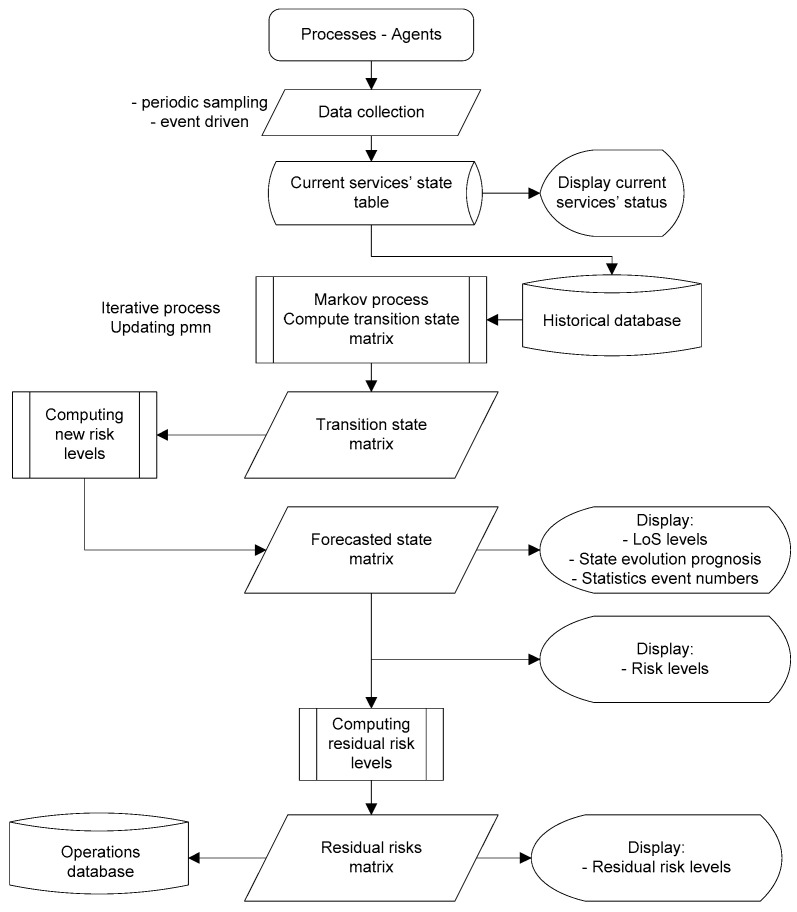
An algorithm for building the forecasted states and residual risks.

**Figure 16 sensors-23-06012-f016:**
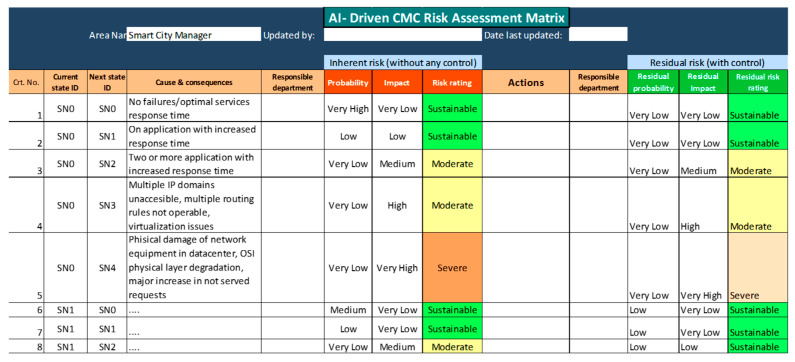
Sample of the AI-Driven Risk Assessment Matrix (AI-DRAM)—screenshot from the application.

**Table 1 sensors-23-06012-t001:** Comparison between different NTMA approaches.

Active Monitoring (Injection of Test Data into the Network)	Passive Monitoring (Big Data Analysis)
Allows for complete end-to-end analysis	Allows for tracing faults in the network
Allows for both asynchronous and synchronous probing of the network (real-time monitoring is possible)	Allows for post-process analysis (non-real time)
Intelligent agents’ usage is possible	Intelligent agents’ integration is possible
Not able to detect clients’ satisfaction	Clients’ satisfaction monitoring is possible
Implementation of self-learning techniques needs maintenance in regard to big data storage	Able to be developed to self-adapting and learning when tracing past events
Oriented more towards Quality-of-Service (QoS)	Oriented more towards Quality-of-Experience

**Table 2 sensors-23-06012-t002:** Generic transition matrix.

State	SN_0_ Normal (100% Operational)	SN_1_ Graceful Degradation	SN_2_ Degradation	SN_3_ Severe Degradation	SN_4_ Catastrophic Failure
SN_0_	pn00	pn01	pn02	pn03	pn04
SN_1_	pn10	pn11	pn12	pn13	pn14
SN_2_	pn20	pn21	pn22	pn23	pn24
SN_3_	pn30	pn31	pn32	pn33	pn34
SN_4_	pn40	pn41	pn42	pn43	pn44

**Table 3 sensors-23-06012-t003:** Availability of services during the test period.

Intelligent Agent	M1	M2	M3	M4	M5	M6	M7	M8	M9	M10	M11	M12	Avg. Val.	Outage Probab.
Traffic	99.65	99.94	99.81	99.88	99.77	99.82	99.95	99.40	99.95	99.38	99.84	99.35	99.73	0.002716667
Energy microgrids	99.64	99.96	99.83	99.74	100.00	100.00	99.95	99.50	99.62	98.82	99.94	99.92	99.74	0.002566667
Environment sensors	99.79	99.76	99.80	99.90	99.95	100.00	100.00	98.75	99.78	99.94	100.00	100.00	99.81	0.001941667
Crowdsourcing	99.81	99.93	99.94	99.93	100.00	100.00	99.97	99.80	99.98	100.00	100.00	99.98	99.95	0.00055
Public lighting	99.64	99.91	99.76	99.62	99.77	99.77	99.90	98.34	99.76	99.88	99.96	99.35	99.64	0.003616667
Waste management	99.24	99.94	99.94	99.85	100.00	99.92	99.92	99.70	100.00	100.00	99.95	99.95	99.87	0.001325

**Table 4 sensors-23-06012-t004:** Numerical example for the analyzed case—the transition matrix.

State Probabilities	SN_0_ Normal (100% Operational)	SN_1_ Graceful Degradation	SN_2_ Degradation	SN_3_ Severe Degradation	SN_4_ Catastrophic Failure
State description	The network of networks is fully operational, all microgrids are operational, and the internet and 5G/LTE are operational	Local sensor/a local data collection network with high response time >10 s	Host domain for one or more services on parent network—non-functional (single IP unreachable)	Several services of the mother network not working (physical equipment in data center faulty—(multiple IPs inaccessible)—use network monitoring tools e.g., PRTG Network Monitor) + warning if data center powered on UPS	Damage of the physical layer in the OSI stack (e.g., FO trunk cut)—major increase in all requests, no service
SN_0_	0.8572	0.0811	0.0352	0.0253	0.0012
SN_1_	0.5114	0.3221	0.1271	0.0382	0.0012
SN_2_	0.2632	0.3200	0.2312	0.1844	0.0012
SN_3_	0.2352	0.3724	0.3459	0.0453	0.0012
SN_4_	0.0253	0.0352	0.0811	0.8542	0.0042

**Table 5 sensors-23-06012-t005:** Predicted state for the analyzed case—a transition matrix after two sampling steps.

State Probabilities	SN_0_ Normal (100% Operational)	SN_1_ Graceful Degradation	SN_2_ Degradation	SN_3_ Severe Degradation	SN_4_ Catastrophic Failure
State description	The network of networks is fully operational, all microgrids are operational, and the internet and 5G/LTE are operational.	Local sensor/A local data collection network with high response time >10 s	Host domain for one or more services on parent network—non-functional (single IP unreachable)	Several services of the mother network not working (physical equipment in data center faulty—(multiple IPs inaccessible)—use network monitoring tools, e.g., PRTG Network Monitor) + warning if data center powered on UPS	Damage of the physical layer in the OSI stack (e.g., FO trunk cut)—major increase in all requests, no service
SN_0_^2^	0.791510479	0.645561482	0.49351611	0.493784618	0.262048035
SN_1_^2^	0.116369461	0.200163222	0.267114672	0.266625072	0.35759367
SN_2_^2^	0.057468856	0.101637221	0.167272171	0.151350561	0.3199232
SN_3_^2^	0.033455707	0.051443922	0.070903142	0.087045872	0.05922247
SN_4_^2^	0.001213629	0.00121363	0.00121363	0.00121363	0.001222558

**Table 6 sensors-23-06012-t006:** *APDX* indexes for all services—one year analysis period, monthly averaged.

Intelligent Agent	M1	M2	M3	M4	M5	M6	M7	M8	M9	M10	M11	M12	Avg. Val.	Insatisfaction Probab.
Traffic	86.00	86.00	90.00	90.00	91.00	90.00	90.00	90.00	90.00	92.00	92.00	92.00	89.92	0.100833333
Energy microgrids	96.00	95.00	94.00	96.00	94.00	96.00	97.00	98.00	93.00	92.00	94.00	98.00	95.25	0.0475
Environment sensors	94.00	94.00	94.00	92.00	92.00	93.00	93.00	93.00	94.00	94.00	96.00	94.00	93.58	0.064166667
Crowdsourcing	93.00	93.00	94.00	93.00	93.00	93.00	95.00	95.00	92.00	88.00	91.00	96.00	93.00	0.07
Public lighting	95.00	95.00	92.00	94.00	95.00	95.00	94.00	94.00	94.00	94.00	94.00	91.00	93.92	0.060833333
Waste management	80.00	77.00	79.00	75.00	77.00	79.00	77.00	76.00	79.00	81.00	81.00	82.00	78.58	0.214166667

**Table 7 sensors-23-06012-t007:** The impacts’ mapping structure.

Impact		Very Low	Low	Medium	High	Very High
Probability	Very High	Sustainable	Moderate	Severe	Critical	Critical
	High	Sustainable	Moderate	Severe	Critical	Critical
	Medium	Sustainable	Moderate	Moderate	Severe	Critical
	Low	Sustainable	Sustainable	Moderate	Severe	Critical
	Very Low	Sustainable	Sustainable	Moderate	Moderate	Severe

**Table 8 sensors-23-06012-t008:** Quantitative assessment of inherited risks.

Impact		Very Low	Low	Medium	High	Very High
Probability	Very High	1				
	High					
	Medium	2	1			
	Low	3	2	1		
	Very Low			2	4	4
	TOTAL	6	3	3	4	4

**Table 9 sensors-23-06012-t009:** Comparison of similar technologies for NTMA.

Classic Approaches Employing DBN/DNN and/or Active Monitoring	Proposed Solution Involving Usage of AI Agents and Passive Monitoring
Complexity in programming	Reduced programming complexity
Needs complex engineering teams	Less demanding in integration
Provides a direct path to failure or faulting application	Only permits post-analysis of failures causes
Not integrating all services and applications	Allows for future state prediction in a certain degree of confidence (confidence may improve in time)

## Data Availability

Data used for this research are unavailable due to privacy.

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
