# Peer review of "Smart Preventive Maintenance of Hybrid Networks and IoT Systems Using Software Sensing and Future State Prediction"

_sensors, 2023, doi:10.3390/s23136012_

Round 1

Reviewer 1 Report

Research on Smart Preventive Maintenance of Hybrid Networks and IoT Systems using Software Sensing and Future State Prediction has attracted lots of interest in recent years. In this research work, a study is carried out to propose a platform for integrated management of preventive maintenance, using software sensing, application monitoring, and Markov-based future state prediction, to increase the resilience of a complex network of networks. Thus, giving an overview of the latest developments and challenges in the field would be of high interest to the scientific community. Unfortunately, at this point, the main features of the manuscript are not presented properly and strong improvements would be needed to make the publication acceptable. In the following, I tried to mention the most crucial shortcomings of the article:

1)      The expression of the Abstract part should be simple and clear, highlighting the key points and main contributions.

2)      The authors included only two research studies based on DBN/DNN in this study. The fact is that there are many research studies exist that have already proposed state-of-the-systems/solutions/models based on deep learning. So please include and discuss those studies briefly in the introduction section (section I) and in detail in the related work section (section 2).

3)      The authors should add a separate section for the related work. The authors need to write this section based on the following comments:

    a.       The first concern is that some recent research papers are missing for discussion in Section 2. Please add some latest research papers in the manuscript.

     b.      In the related section (section 2), the authors must mention the limits of existing approaches. For a better understanding, I advise the authors to discuss the merits and demerits in tabular form and give your critical analysis.

4)      The description of the proposed methodology in Section II is too abstract and generic. The novelty of the proposed work must be highlighted in this Section. At the start of the section, a brief overview should be provided. How proposed method works in the context of this study?

5)      In the results section (Section 3), add an explanation about why the proposed methods achieved better results based on what parameters. The authors must discuss the computational overhead in the cost and complexity of the research studies.

6)      In section 4 (Discussion), the authors must present the comparative result and analysis in a separate table based on the proposed work and existing research studies using different approaches.

7)      I also suggest briefly describing or adding some futuristic applications of the methods which employ state-of-the-art techniques.

8)      The limitations and challenges must also be summarized before the conclusion section.

9)      In Conclusion – it seems it is just a recap of basic concepts. Please be clear in writing the conclusion with facts, evidence, or numerical values if any.

10)  The quality of Figures # 1 to 15 is poor. I suggest using a vector format like eps or .png format to insert images.

11)  Figures # 16 and 17 are not figures, these are the Tables. Please re-arrange the Tables and Figures after making the necessary changes.

1) The manuscript requires English proofreading. It contains many grammar errors, punctuation errors, white spaces, and typos. It is suggested that the author polish the manuscript carefully to enhance its readability.

Author Response

Many thanks for your comprehensive and useful review! The article has been submitted to a thorough process of revision and completion. Please find our responses to your requests, in the attached document.

Reviewer 2 Report

Abstract: Need Revision

Keywords: Good

Introduction: Average/Revised Please

Literature Review: Average

Data Set: Average

Methodology: Weak

Caption, Citations & Footnotes: Good

Pictures, graphs & Flowcharts: Average

Results: Weak

Conclusion: Average

Future Work: Poor

References: Average

----------- Overall evaluation -----------

Suggestion and Recommendation:

1. In the introduction, the scientific problem of the existing evaluation is missing. There should initially be discussed the actual problem and then the research motivation.

2. Please highlight major contributions of this work in this current version, otherwise the current form shows weak/lack of novelty.

3. Please refine the language of this paper, such as avoid we, they, our, and other related words in this paper.

4. Check mathematical description in the proposed work section, the expression showing totally general, no customize or testing samples are discussed/presented.

5. Authors are encouraged to base on recent references about the current development in blockchain technology. Moreover, technology collaborates with other technologies to create new paradigms, such as artificial intelligence, such machine learning, deep learning, with federated learning.

6. Please improve the portion of problem description and problem formulation of the proposed work. Cannot find novelty in the current form.

7. In methodology, only a single algorithm presented to demonstrate the working operation? authors should be clarifying the events of execution (one-by-one) for the whole process? Please elaborate. Also, the experiments should be expanded including more analysis and comparisons with other indexes/baselines and compare your proposed method with newly state-of-the-art methods.

8. The topic is very good and unique but need to improve paper organization.

Minor English Language Editing is Required.

Author Response

Many thanks for your comprehensive and useful observations! The article has been submitted to a thorough process of revision and completion. Please find our answers in the attached file.

Reviewer 3 Report

- Line 197: \lambda(t) shoudl be italic.

- The algorithm for building the transition matrix in Fig. 8 is not celalry explained. It is not clear how the main loop for building the transition matrix works.

. Fig. 15-16: please discuss the relation between the mapping of impact degrees and temporal features.

Author Response

(The authors gave the same response as above.)

Reviewer 4 Report

This work proposes an automated tool to assist maintenance operations for complex, heterogeneous systems, and data communication networks. It offers a detailed description of the proposed platform and it is based on authors previous work [38]. However, the manuscript could be improved considering the following

1.In introduction the authors mention that they propose an integrated platform for preventive maintenance dedicated to complex smart city services data communication networks, but they did not compare their work with other risk assessment platforms and they do not include methodologies for risk assessment like analytical models that they used to investigate the network performance for all key performance indicators([1]).

2.In section 3 it is not clear how the authors Build a Markov Chain Model for Risk Prediction. Moreover figure 15 does not depicts how the transition probability matrix is generated.

3.In discussion the authors do not compare the  AI-DRAM algorithm with other optimization algorithms based on Markov chains ([3]-[4]) and other solutions [2]

[1] A. U. Khan, G. Abbas, Z. H. Abbas, M. Bilal, S. C. Shah and H. Song, "Reliability Analysis of Cognitive Radio Networks With Reserved Spectrum for 6G-IoT," in IEEE Transactions on Network and Service Management, vol. 19, no. 3, pp. 2726-2737, Sept. 2022, doi: 10.1109/TNSM.2022.3168669.

[2]Zahid, N., Sodhro, A. H., Kamboh, U. R., Alkhayyat, A., & Wang, L. (2022). AI-driven adaptive reliable and sustainable approach for internet of things enabled healthcare system. Math. Biosci. Eng, 19, 3953-3971.

[3]Peiravi, A., Nourelfath, M., & Zanjani, M. K. (2022). Redundancy strategies assessment and optimization of k-out-of-n systems based on Markov chains and genetic algorithms. Reliability Engineering & System Safety, 221, 108277.

[4]Kattepur, A., Nair, A. R., Saimler, M., & Donmez, Y. (2022, September). Industrial 5G Service Quality Assurance via Markov Decision Process Mapping. In 2022 IEEE 27th International Conference on Emerging Technologies and Factory Automation (ETFA) (pp. 1-8). IEEE.

Author Response

(The authors gave the same response as above.)

Round 2

Reviewer 1 Report

I recommend accepting the paper for publication as it addresses all my previous comments supported by the comprehensive revision and adds significant value to their revised work.

Author Response

Thank you for your useful observations and recommendations!

Reviewer 2 Report

The author of this paper addressed all my concerns.

Please accept this current version.

Thanks for your effort.

Acceptable.

Author Response

(The authors gave the same response as above.)

Reviewer 4 Report

1. According point A1 the authors states that the presented work is an additional service for improving the response time of revealing failures this is not clarified in introduction..

2.Point A2.

Based to your manuscript you have

              Identify the different states or categories that describe the risk levels

              Gather data- information on the transitions between states.

              Count transitions

              And finally you Calculate probabilities and you Construct the matrix

 Since the authors do not introduce a new Hidden Markov model for reliability then it is difficult for someone to figure out the contribution of the presented work.

The authors should include in discussion section two paragraphs for explaining the contribution of their work provide and they have to include the benefits among other similar algorithms in the literature [1]

[1] María Luz Gámiz, Nikolaos Limnios, María del Carmen Segovia-García,Hidden markov models in reliability and maintenance,European Journal of Operational Research,Volume 304, Issue 3,

2023,Pages 1242-1255,ISSN 0377-2217,

https://doi.org/10.1016/j.ejor.2022.05.006.

Author Response

Thank you for your useful observations and recommendations! Please find attached our answers. 
